# Research on Performance Deterioration of Multi-Walled Carbon Nanotube–Lithium Slag Concrete under the Coupling Effect of Sulfate Attack and Dry–Wet Cycles

**DOI:** 10.3390/ma16145130

**Published:** 2023-07-20

**Authors:** Yifei Zhang, Yongjun Qin, Zheyi Guo, Dongjin Li

**Affiliations:** 1School of Civil Engineering and Architecture, Xinjiang University, Urumqi 830047, China; iris@stu.xju.edu.cn (Y.Z.); 875715741@stu.xju.edu.cn (Z.G.); 107552104191@stu.xju.edu.cn (D.L.); 2Xin Jiang Key Lab of Building Structure and Earthquake Resistance, Xinjiang University, Urumqi 830047, China

**Keywords:** MWCNTs concrete, dry–wet cycle, sulfate attack, pore deterioration, X-CT scanning

## Abstract

Sulfate attack is one of the main factors affecting the durability of concrete structures. In recent years, multi-walled carbon nanotubes (MWCNTs) have attracted the attention of scholars for their excellent mechanical properties and durability performance. In this paper, the influence of sulfate attack and dry–wet cycles on the performance of multi-walled carbon nanotube–lithium slag concrete (MWCNT-LSC) with varied MWCNT content (0 wt.%, 0.05 wt.%, 0.10 wt.%, and 0.15 wt.%) and varied water–cement ratios (0.35, 0.40, and 0.45) were investigated. In addition, scanning electron microscopy (SEM) and X-ray computed tomography (CT) tests were conducted to analyze the microstructure and pore structure of the concrete. The results showed that concrete incorporated with MWCNTs could effectively mitigate sulfate attack. The resistance to sulfate attack of concrete was negatively related to the water–cement ratio when the dry–wet cycle was fixed. The MWCNT-LSC showed the best compressive strength at the water–cement ratio of 0.35 and 0.10 wt.% MWCNTs. The SEM test results showed that the MWCNTs filled the pores and cracks within the specimen and formed bridges between the cracks, enhancing the resistance to sulfate attack. The CT test results also showed that the addition of MWCNTs could reduce the porosity of concrete, refine the pore size and inhibit the generation and development of cracks, thus optimizing the internal structure of concrete and improving its resistance to sulfate attack.

## 1. Introduction

The service life of building structures is governed by the problem of concrete durability. Aggressive salts in the environment are a significant cause of the deterioration of concrete properties. In particular, sulfate attack is one of the main factors affecting the durability of concrete [1,2,3,4,5,6]. Sulfate ions can cause spalling of the concrete cover, corrosion of the internal reinforcement, and a reduction in load-bearing capacity, thus shortening the service life of the concrete structure. In addition, these damages are also accompanied by significant maintenance and protection costs. When exposed to a dry–wet cycle, concrete structures deteriorate at a rate significantly higher than the rate of deterioration observed when specimens are submerged for an extended period [7]. Sulfate not only reacts with hydration products in concrete to produce expansive calcium alumina and gypsum but also causes Ca^2+^ dissolving, resulting in the deterioration of concrete properties [8,9,10,11].

Nanoparticles of appropriate size can fill the cementitious pores of cement hydration products and reduce the amount of cement-based materials [12]. This could be attributed to the properties of the small particles and the high surface-area-to-volume ratio of nanomaterials. In addition, the microstructures of cement-based materials could be optimized, the bonding strength between cement and aggregates could be improved, and the flaws of the internal cement-based materials could be addressed [13,14]. The discovery of carbon nanotubes (CNTs) in 1991 was considered one of the most important discoveries in the field of materials [15]. Carbon nanotubes come in two forms, single-walled carbon nanotubes (SWCNTs) and multi-walled carbon nanotubes (MWCNTs). Both are well-structured graphene cylinders with an aspect ratio (the ratio of the length to the circumference of the cross-section) of 1000 or more [16,17,18,19]. Therefore, carbon nanotubes are ideal for reinforcing composite materials [20]. MWCNTs are nanomaterials that exhibit excellent performance. MWCNTs significantly influence the macroscopic and microstructural characteristics of cement-based materials [21,22,23,24]. In addition, MWCNTs can improve the porosity and pore size distribution of cement-based materials by exploiting nano-filling and nucleating functions [25,26,27]. MWCNTs can also strengthen the internal interface transition zone of cement-based materials and significantly lower the production of microcracks by exercising bridging effects [28]. In addition to its composition, the mesoscopic structure of concrete also affects its performance. The fundamental component of the mesoscopic structure is the pore structure, which is strongly related to the macroscopic qualities of concrete, such as its ability to resist sulfate attack.

Research on carbon nanotube–cement matrix composites is still exploratory [29,30]. Scholars have found that the mechanical properties of carbon nanotube–cement matrix composites were closely related to the content of carbon nanotubes. Liu et al. [31] investigated the effect of the optimum mix proportion and salt freezing durability of MWCNT UHPC with different MWCNT contents and water binder ratios. The results showed that the optimum mix proportion was at a 0.19 water binder ratio and 0.1% carbon nanotube content. Souza et al. [32] found that multi-walled carbon nanotubes increased the compressive strength and reduced the rheological properties of Portland cement repair mortar to some extent. Xu et al. [33] conducted a coupling test of sulfate attack and freeze–thaw cycles on multi-walled carbon nanotube–fly ash concrete (MWCNT-FAC). The results showed that the best improvement in the salt freeze–thaw resistance of MWCNT-FAC was achieved when the MWCNT content was 0.05 wt.%. The experimental results of Wang et al. [34] showed that the addition of MWCNTs can improve the chloride ion penetration and sulfate attack resistance of cement/MWCNTs composites. Gao et al. [35] found that the incorporation of MWCNTs improved the freeze–thaw resistance of concrete under freeze–thaw and sulfate attack environments. Alicia et al. [36] found that the addition of CNTs increased the rate of hydration of the specimens, leading to an acceleration of microstructure formation. Alastair et al. [37] found that the optimal CNT content for enhancing resistance to chloride diffusion is 0.05–0.1 wt.%. Hawreen et al. [38] analyzed the effect of concrete reinforced with different weight fractions of carbon nanotubes. The results showed that using 0.1 wt.% MWCNTs increased the compressive strength by about 20%. According to Hu et al. [39], the ideal MWCNT content is 0.1% of the cement mass, and under these dosage conditions, the porosity of the cement paste can be decreased by 27.52%.

Studies on the effect of sulfate attack on MWCNTs concrete were still minimal. This paper investigated the effects of multi-walled carbon nanotubes (MWCNTs) on the mechanical properties and microstructure of lithium slag concrete (LSC) during sulfate attack and drying–wetting cycles. The deterioration pattern of lithium slag concrete under sulfate attack was investigated by studying the changes in morphology, compressive strength, and mass loss of multi-walled carbon nanotube–lithium slag concrete. The microstructure of multi-walled carbon nanotube–lithium slag concrete (MWCNT-LSC) was analyzed by scanning electron microscopy (SEM). The internal porosity, pore size distribution, sphericity, and compactness were examined using X-ray computed tomography (CT) images.

## 2. Materials and Methods

### 2.1. Materials

The cementitious material used in this study was a composite of P⋅O 42.5 ordinary Portland cement (OPC) and lithium slag (LS), which was taken from the Urumqi lithium salt plant, dried and ground, and put into the experiment with a density of 2.48 g/cm^3^ and a specific surface area of 417 m^2^/kg. The main chemical composition of cementitious material is shown in Table 1. The fine aggregate used in this study was river sand with an apparent density of 2641 kg/m^3^ and a fineness modulus of 2.8. The coarse aggregate was continuously graded gravel with a 5–20 mm particle size and an apparent density of 2687 kg/m^3^. The water-reducing agent was a high-performance polycarboxylic acid water-reducing agent. The test water was taken from municipal tap water. The multi-walled carbon nanotubes chosen for this experiment were prepared by Chengdu Jiacai Technology Co., Ltd. (Chengdu, China), following the CVD process. The physical parameters are shown in Table 2. The sulfate solution was prepared from 5% anhydrous sodium sulfate using a mass fraction.

### 2.2. Specimen Preparation

In total, 12 groups of MWCNT-LSC with different specimens were prepared, where the water–cement ratio (W/C) was varied as 0.35, 0.40, and 0.45, and the content of MWCNTs was 0 wt.%, 0.05 wt.%, 0.10 wt.%, and 0.15 wt.% of the cement mass; the mix proportions are shown in Table 3, where 35, 40, and 45 represent water–cement ratios of 0.35, 0.40, and 0.45 and 0, 5, 10, and 15 represent MWCNT doses of 0 wt.%, 0.05 wt.%, 0.10 wt.%, and 0.15 wt.%. According to GB/T 50080-2016 [40], the materials were placed in a 100 mm × 100 mm × 100 mm cube mold and shaken vigorously on a vibrating table. The specimens were maintained at 20 ± 2 °C and relative humidity ≥95%.

### 2.3. Experiment Methods

#### 2.3.1. Coupling of Sulfate Attack and Drying–Wetting Cycles Test

According to GB/T 50082-2009 [41], the concrete specimens were cured under standard conditions for 26 d and then taken out from the standard curing room; the surface moisture was dried, and the specimens were put into an oven and baked at 80 ± 5 °C for 48 h, cooled to room temperature, and then put into the sulfate test chamber with 5% sodium sulfate solution for dry–wet cycle test. A CABR-LSB automatic concrete sulfate dry–wet cycle test machine was used for the dry–wet cycle experiment. The specimens were immersed in sulfate solution for 15 h, air-dried for 1 h, baked for 6 h at 80 ± 5 °C, and cooled for 2 h. The duration of one wet and dry cycle was 24 h, with a total of 0, 30, 60, 90, 120, and 150 dry–wet cycles. During the experiment, the pH was maintained in the range of 6–8, and the pH of the solution was tested every 15 cycles. The specimen was removed for drying after reaching the set number of dry–wet cycles.

#### 2.3.2. Strength and Mass Loss Rate Test

The specimens were tested for compressive strength and quality at the end of every 30 dry–wet cycles. According to standard GB/T 50081-2019 [42], the compressive strength was conducted using a YAW 2000 A electro-hydraulic servo pressure testing machine (Shanghai Bairoe Testing Instrument Co., Shanghai, China). A total load of 0.5 MPa/s was applied. The acquisition system logged the peak load during the test. Three specimens were tested for each group, and the average value was calculated. The concrete compressive strength fcc (MPa) and mass loss rate (%), were calculated using Equations (1) and (2):(1)$$fcc=\frac{P}{A}$$
where *P* (N) is the peak load and *A* (mm^2^) is the pressure area of the specimen.
(2)$$\Delta W=\frac{G_{0}-G_{i}}{G_{0}}\times 100\%$$
where Δ*W* (%) is the mass loss rate of concrete after erosion, *G*_0_ (g) is the initial mass of concrete before erosion, and *G_i_* (g) is the *i*-th measured mass of concrete after erosion.

#### 2.3.3. Microstructure Test

An SU8010 scanning electron microscope from Hitachi, Tokyo, Japan, was used to observe the microstructure of MWCNT-LSC at different numbers of dry–wet cycles. The specimen size of the concrete test was about 8 mm, and the non-observed surface was polished and glued to the conductive glue of the disc, followed by gold spraying.

The pore structures of the MWCNT-LSC specimens were investigated using X-ray computed tomography technology (CT). The concrete utilized in this CT test was scanned using Y.CT COMPACT equipment (YXLON, Hamburg, Germany, see Figure 1). The scanning parameters are presented in Table 4. Two elements dictate the performance of the fan-beam ICT scanning equipment: the line detector and the high-energy X-ray tube. During operation, the turntable and high-dose rays travel vertically in tandem with the rotating components, and an image is produced using attenuation data. The CT images were pre-processed using MATLAB tool functions and the IMAGE J V 1.8.0.112 machine learning plug-in. Image enhancement, denoising, and morphological image processing were used to achieve the goal of image improvement and analysis. Two-dimensional pore characteristic parameters such as pore size and surface area were extracted using Image Proplus. Then, these parameters were further analyzed by using the mean and maximum values as indicators. The specimens were rebuilt using VGStudio to build three-dimensional pore structure models, as shown in Figure 2.

## 3. Experimental Results and Discussion

### 3.1. Morphology Change

The specimens were observed after every 30 sulfate dry–wet cycles. The macroscopic morphology of specimens after 30, 60, 90, 120, and 150 sulfate dry–wet cycles is shown in Figure 3. Concrete surface cracks have been indicated by the red rectangles in Figure 3. After 30 and 60 sulfate dry–wet cycles, some cracks were observed on the surfaces of the specimens, but there was no significant damage. However, some specimens had small pores due to the manufacturing method. After 30 sulfate dry–wet cycles, it can be observed that specimen 35-0 had more cracks than specimens 35-5, 35-10, and 35-15. After 60 sulfate dry–wet cycles, more cracks were observed in specimen 45-0 than in specimens 45-10 and 45-15. The addition of MWCNTs seemed to limit the deterioration of the concrete, as the specimens with MWCNTs had fewer cracks than those without MWCNTs. After 90 sulfate dry–wet cycles, the specimens were relatively intact in appearance. Two cracks with a width of about 1.5–2 mm were observed in the upper left and right corners of specimen 45-5. Similarly, a penetrating crack was observed with a width of about 1.5–2 mm after 120 and 150 sulfate dry–wet cycles. This was because the erosion reaction between sulfate ions and cement hydration products generates expansion erosion products, such as gypsum and AFt [43], causing subsequent cracking, surface spalling, and concrete spalling. After 150 sulfate dry–wet cycles, all specimens exhibited a large number of pores. The LSC with 0.10 wt.% MWCNTs exhibited less severe damage, and few crystals were collected on the specimen surface. Fewer cracks and low corner dropouts were observed in these conditions. The larger W/C made the transport faster, and the accumulation of sulfate ions in MWCNT-LSC was expected to be responsible for more serious damage to the exposed concrete surface. The specimen with MWCNTs had fewer cracks than specimen 35-0 under the W/C of 0.35. In the specimens with the W/C of 0.45, visible cracks and observed dross and corners, as well as leakage of aggregate particles, can be seen. In contrast, no significant damage was observed for the other specimens with lower W/C. Overall, the addition of 0.10 wt.% of MWCNTs improved the resistance of the concrete specimens to sulfate attack.

### 3.2. Compressive Strength

The compressive strength of MWCNT-LSC after 0, 30, 60, 90, 120, and 150 sulfate dry–wet cycles is presented in Figure 4 and Appendix A, Table A1. Figure 4 clearly shows that with the increase in MWCNT dosing, the compressive strength of the specimens first increased and then decreased, which was in agreement with the results reported by references [31,44]. The initial compressive strength of specimen 35-10 was the maximum, recorded as 71.3 MPa. At the same W/C, the compressive strength of the specimens containing MWCNTs was greater than that of the specimens without MWCNTs. As the number of sulfate dry–wet cycles rose, the compressive strength of the concrete also started to increase.

The compressive strength, recorded in the presence and absence of MWCNTs and at different W/C conditions, increased significantly after 30 sulfate dry–wet cycles, where the maximum growth rate was 35.67%. During this period, sulfate ions entered the concrete and produced calcium alumina, which refilled the pore structure of the concrete and effectively reduced the porosity [31], thus allowing the compressive strength of the concrete to increase.

After 30 and 60 sulfate dry–wet cycles, as the number of cycles rose, the compressive strength of the part of the specimens kept rising while others started to decrease. The compressive strength of all specimens with 0.10 wt.% MWCNTs continued to increase. The filling of the pores at the interior of the specimens with MWCNTs might cause this phenomenon. This helped improve the pore structure of concrete, promote hydration, and increase the compactness of specimens [44]. MWCNTs may also fill the internal pores of concrete because of their small size. Additionally, the bridging effect could limit the generation of corrosion products, hinder the development of microcracks, and prevent the deterioration of the compressive strength.

The compressive strengths of all the specimens began to decrease after 60 sulfate dry–wet cycles due to the increase in the extent of corrosion at this point. The pores were gradually filled, altering the internal structure of the concrete and increasing its compactness. SO_4_^2−^ reacted with calcium hydroxide and hydrated calcium aluminate to form ettringite, and the solid phase volume increased, which resulted in the generation of a large amount of expansion stress inside the concrete. However, due to the addition of MWCNTs, sulfate had little effect on LSC.

### 3.3. Mass Loss Rate

The mass loss rate of MWCNT-LSC after 30, 60, 90, 120, and 150 sulfate dry–wet cycles is presented in Figure 5 and Appendix A, Table A2. Figure 5 shows the mass loss rate of each specimen under various sulfate dry–wet cycling. The mass loss rate increased at the beginning of the test and then decreased as the number of cycles increased. The mass loss rate was negative for all specimens until 90 sulfate dry–wet cycles, indicating that the quality of the concrete increased with the number of sulfate dry–wet cycles. This phenomenon occurred due to the chemical reaction between the salt crystallization and the salt ions inside the concrete. Also, the evaporation of the solution from the surface of the concrete facilitated this process. Salt solution crystallization was attached to the surface of the specimen. After 90 sulfate dry–wet cycles, the mass loss rate of most specimens began to rise due to the filling effects of the sulfate crystals and the early generation of expansive sulfate erosion products (e.g., AFt and gypsum) [45,46,47]. This expansion led to fractures and the widening of cracks, causing the aggregate to flake off and a deterioration in the quality of the specimens. The density of the concrete decreased due to a reduction in the content of the calcium silicate hydrate gel. After 150 sulfate dry–wet cycles, the mass loss rates were all positive, indicating that the quality of the concrete was decreasing rapidly. This is mainly due to the erosion of the specimens under the action of sulfate, which results in the expansion pressure being greater than the cementing force between the materials. It is clear that regardless of the amount of MWCNTs added, specimens with a W/C of 0.45 had more significant mass loss than those with a W/C of 0.40. Specimen 45-10 was looser and had greater mass variation compared to those with a W/C value of 0.35. This indicated that a large amount of sulfate had the potential to penetrate the interior of the concrete, where it precipitated as sulfate crystals, increasing the mass of the concrete. At the same W/C conditions, the mass variation was more pronounced for the specimens without MWCNTs than for those with MWCNTs. The other experimental groups with MWCNTs showed a slow increase in mass loss rate throughout the process. This was not the case for specimens 45-5 and 45-15, which were significantly disrupted.

### 3.4. Scanning Electron Microscope Observation

The microstructures of MWCNT-LSC under sulfate dry–wet cycling are shown in Figure 6. It can be seen from Figure 6 that most of the multi-walled carbon nanotubes were bridged with the hydration products in the form of single laps, forming multiphase composites and improving the mechanical properties of the concrete. MWCNTs could not only fill the pores and cracks within the specimens and delay or even hinder the penetration and diffusion of destructive sulfate ions, but also form bridges between the cracks to hinder their expansion. As can be seen from the microstructure of the concrete before the sulfate dry–wet cycling, unhydrated LS particles were present inside the specimens without MWCNTs, while for the specimens with MWCNTs, the MWCNTs were dispersed between the cracks within the concrete specimens and bridged, acting as a strong reinforcement. As the W/C increased, more cracks and pores appeared in the internal structure, but it was evident that MWCNTs formed bridges between the cracks. After 150 sulfate dry–wet cycles, the hydration products of specimens 35-10 and 45-10 were still relatively dense, with no significant loosening or prominent porosity. This phenomenon indicated that the specimens with MWCNTs have a long-lasting hydration and bridging effect, which made the concrete denser internally and thus effectively improved the sulfate resistance of MWCNT-LSC.

### 3.5. Evolution Rules of Concrete Microstructure

#### 3.5.1. Two-Dimensional Pore Structure Analysis

The interval between the CT scanning stages should be minimized to achieve high accuracy in subsequent modeling stages. This resulted in generating a very high number of CT scanning images of a single specimen. The pore structure was retrieved and evaluated by Image Proplus after 100 CT images were chosen at equal intervals along the scanning z-axis during the 2D study. Alternatively, a representative scanning segment was chosen every 1 mm. The fluctuation in the 2D porosity of the specimens along the depth of the *z*-axis is depicted in Figure 7. It was clear that each specimen’s variation in 2D porosity changed with changes in the depth of the specimen. The porosity on the upper surface was in the range of 40–70 mm, and the value approached the average value.

Based on the two-dimensional porosity image shown in Figure 7a–f, it was observed that at the same water–cement ratio and in the absence of sulfate attack, the porosity of the LSC without MWCNTs was lower than that of the specimens with MWCNTs. This was because MWCNTs made the cement mortar denser and more difficult to vibrate. However, the pores of the concrete became larger after vibrating. Therefore, the slurry with MWCNTs should be vibrated vigorously to improve the performance of MWCNTs. In contrast, the porosity of the LSC specimen without MWCNTs steadily increased as the number of sulfate dry–wet cycles rose. Meanwhile, the porosity of the LSC with MWCNTs decreased after 30 and 150 sulfate dry–wet cycles, after which the average surface area increased by 11.79% and 35.15%, respectively. The incorporation of MWCNTs effectively prevented the degeneration of pores, indicating that as the number of sulfate dry–wet cycles increased, the MWCNT concrete became significantly more resistant to sulfate attack.

The comparison of the two-dimensional porosity of specimen 35-10 and specimen 45-10 is presented in Figure 7d–i. It was observed that the porosity of the specimen was low when the W/C was relatively small. When the W/C was high, the amount of cement was reduced, resulting in a lower cement paste concentration. Therefore, the specimens could not firmly adhere to the surface of the aggregate or effectively wrap the aggregate. This led to a reduction in the bonding area between the slurry and the aggregate, resulting in the formation of a large number of pores, which reduced the density after hardening. For specimen 45-10, the porosity decreased by 11.49% after 30 sulfate dry–wet cycles. In comparison, it increased by 13.86% after 150 sulfate dry–wet cycles, indicating that MWCNTs effectively delayed the early stages of pore degradation. As sulfate dry–wet cycling increased, the amount of corrosion products in high W/C concrete increased, causing destructive expansion and the formation of more porous and cracked surfaces.

Two-dimensional porosity is characterized by changes in pore size and surface area [48]. The average and maximum pore sizes are shown in Table 5. The average surface area of the specimen without MWCNTs (35-0) increased as the number of sulfate dry–wet cycles increased. After 30 and 150 sulfate dry–wet cycles, for the 35-0 specimen, the average pore size increased by 7.17% and 13.95%, the maximum pore size increased by 22.82% and 15.34%, and the average surface area increased by 11.79% and 35.15%, respectively. Wide cracks and pores were formed, and the specimens were severely damaged, suggesting that a large amount of sulfate entered the specimens. This indicated that MWCNTs played an inhibitory role in pore degradation. There was an initial decrease and subsequent increase in porosity in specimens with MWCNTs and a high W/C. However, the average pore size, maximum pore size, and average surface area of specimens (35-10) containing MWCNTs and with a low water–cement ratio showed a downward trend. After 30 and 150 sulfate dry–wet cycles, the average pore size decreased by 0.60% and 3.87%, the maximum pore size decreased by 11.85% and 2.48%, and the average surface area decreased by 2.72% and 12.46%, respectively, further indicating that MWCNTs played an inhibiting role in pore degradation.

#### 3.5.2. Three-Dimensional Pore Structure Analysis

A 3D pore feature model was built using the VGStudio superimposition algorithm to visualize the natural morphology of the pores, which facilitated the evaluation of the morphological changes of the pores. In order to evaluate the overall porosity in the specimen, 3D pore characteristics were calculated, and the results are shown in Figure 8 and Figure 9.

It can be seen from Figure 8 that the porosity of the specimens without MWCNTs grew with the number of sulfate dry–wet cycles, as shown by an increase of 31.84% and 18.83% after 30 and 150 sulfate dry–wet cycles, respectively. After 150 sulfate dry–wet cycles, the porosity of specimen 35-10 decreased by 9.34%, while the porosity of specimen 45-10 increased by 55.19%. The 3D pore characterization results further confirmed the inhibitory effect of MWCNTs and low W/C on pore deterioration under sulfate dry–wet cycling. This was similar to the result of 2D pore analysis.

#### 3.5.3. Frequency Distribution of Pore Size

The characterizations of pore size and distribution were complicated. For this section, the pore sizes were graded from 0 based on the statistical results. The pore diameter distribution after different numbers of sulfate dry–wet cycles is shown in Figure 10 and Table 6.

The pores are classified into four types according to the pore size [49]: small pores (<0.3 mm diameter), medium pores (a diameter of 0.3–0.6 mm), large pores (a diameter of 0.6–0.9 mm), and oversized pores (a diameter greater than 0.9 mm). As shown in Figure 10, when the 35-0 specimens were not eroded, the pore size ranged from 0 to 0.3 mm. As the number of sulfate dry–wet cycles increased, the percentage of pores in the 0~0.3 mm range decreased, while the percentage of pores larger than 0.3 mm increased. It was observed that after 30 sulfate dry–wet cycles, the percentage of 0~3 mm pores in specimen 35-0 was 11.18% lower relative to the uncorroded condition. After 150 sulfate dry–wet cycles, most of the pores in the specimens were 0.3~0.6 mm, and the proportion of pores larger than 0.9 mm increased. In addition, there was no evidence of the 5.4~5.7mm and 5.7~6 mm pores prior to erosion. Bigger pores were formed under sulfate dry–wet cycling, destroying the pore structure. When specimen 35–10 was not eroded, the majority of the pore sizes were in the range of 0.3~0.6 mm. As the sulfate dry–wet cycling increased, the proportion of the 0~0.3 mm pores did not vary. The content of the 0.3~0.6 mm pores decreased by 0.94% after 30 sulfate dry–wet cycles and expanded by 5.17% after 150 sulfate dry–wet cycles. The percentage of pore sizes larger than 6 mm dropped. The fraction of small pores barely changed, whereas the proportion of large pores decreased, revealing that the MWCNTs effectively attenuated the degradation of pores. Most pores in specimen 45–10 were between 0.3mm and 0.6 mm in size. Nevertheless, after 30 and 150 sulfate dry–wet cycles, the number of pores in this range decreased by 8.27% while the fraction between 0 and 0.3 mm increased. The percentages of pores of 5.4~5.7 mm, 5.7~6 mm, and 6~6.3 mm were reduced by 26.55%, 75.51%, and 63.26%, respectively. MWCNTs can improve the internal pore structure of concrete since MWCNTs act as fillers to fill pores and cracks and as fibers to bridge cracks [50]. It is evident that large pores appeared for a higher W/C after 150 sulfate dry–wet cycles, and the addition of MWCNTs served further to optimize the internal pore structure of the concrete, making it denser inside. This indicates that MWCNTs can indeed mitigate the pore degradation of LSC by optimizing the pore size distribution.

#### 3.5.4. Compactness and Sphericity

The development of the 3D model of the pores enabled the visualization of the true form of the pores, and the actual shape of the pores can be described using two 3D metrics: compactness and sphericity. The specific calculation was based on Equations (3) and (4). The results are shown in Figure 11 and Figure 12.
(3)$$Compactness=\frac{V_{defect}}{V_{sphere}}$$
where *V_defect_* is three-dimensional pore volume and *V_sphere_* is external sphere volume.
(4)$$\mathrm{Sphericity}=\frac{\pi^{\frac{1}{3}}{\left(6\times V\right)}^{\frac{2}{3}}}{A}$$
where *V* is the three-dimensional pore volume and *A* is the three-dimensional pore surface area.

The compactness and sphericity were used to measure the regularity of the pores. The closer the two values were to 1, the greater the resemblance of the pore with a regular circle. According to Figure 12, the average compactness of the specimens without MWCNTs was lower than that of the specimens with MWCNTs, while the compactness of the specimen decreased with the increase in W/C. This indicated that the specimens with MWCNTs and low W/C exhibited better pore characteristics.

Figure 12 shows that the frequency increased as the sphericity increased, while it decreased with increasing sphericity after the sphericity reached approximately 0.6. The sphericity of most pores was between 0.4 and 0.7. Only that of a few pores was above 0.8, which suggested that the closer the range of standard circles, the smaller the number of pores. More than half of the specimens had a sphericity greater than 0.6, demonstrating that most specimens had a pore form closer to the ideal sphere. In addition, it could be observed that the total frequency of specimen 35-10 exceeded 60% when the sphericity was between 0.4 and 0.7, which was, on average 12.27% higher than that of specimens without MWCNTs, and 27.25%, 6.19%, and 5.11% higher after 0, 30, and 150 sulfate dry–wet cycles, respectively. This further showed that the addition of MWCNTs had a positive effect on pore structure optimization.

## 4. Conclusions

This study studied the mechanical and durability properties of multi-walled carbon nanotube–lithium slag concrete (MWCNT-LSC) after sulfate dry–wet cycles. Microstructure and pore structure were analyzed based on SEM and CT test results. The main conclusions are summarized as follows:

(1) MWCNTs significantly increased the compressive strength of LSC. The compressive strength increased and then decreased with the increase in MWCNT content. The LSC compressive strength reached the maximum value of 71.3 MPa when the water–cement ratio was 0.35 and the MWCNT content was 0.10 wt.%. Compared with the specimens without MWCNTs, the compressive strength increased by 4.7%.

(2) A moderate amount of MWCNTs enhanced LSC resistance to sulfate attack after sulfate dry–wet cycles. After 150 sulfate dry–wet cycles, most of the MWCNT-LSC specimens showed no significant changes in appearance. The compressive strength tended to increase and decrease as the number of sulfate dry–wet cycles increased, but the mass loss rate tended to decrease and then increase.

(3) An SEM test showed that MWCNTs improved the resistance of concrete to sulfate attack by filling pores and forming bridges in cracks within the concrete, hindering crack expansion.

(4) A CT test showed that the addition of MWCNTs could effectively reduce the porosity of the samples. In addition, MWCNTs could optimize the pores and improve the sphericity and compactness.

(5) The above conclusions provide references for further research on MWCNT-LSC and its potential applications in construction. In addition, it should be noted that more in-depth and refined studies on MWCNT-LSC exposed to different aggressive environments are required in the future.

## Figures and Tables

**Figure 1 materials-16-05130-f001:**
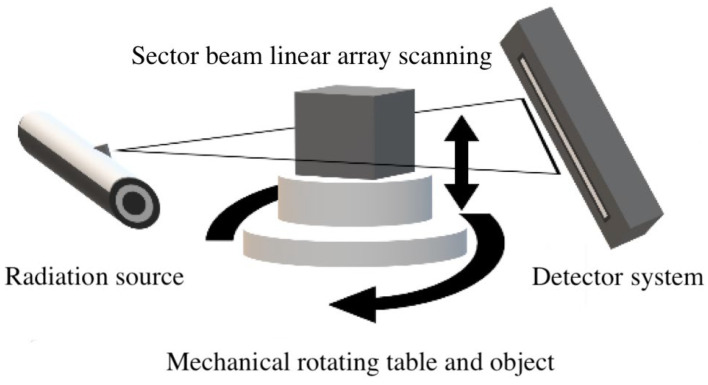
Industrial CT scanning principle.

**Figure 2 materials-16-05130-f002:**
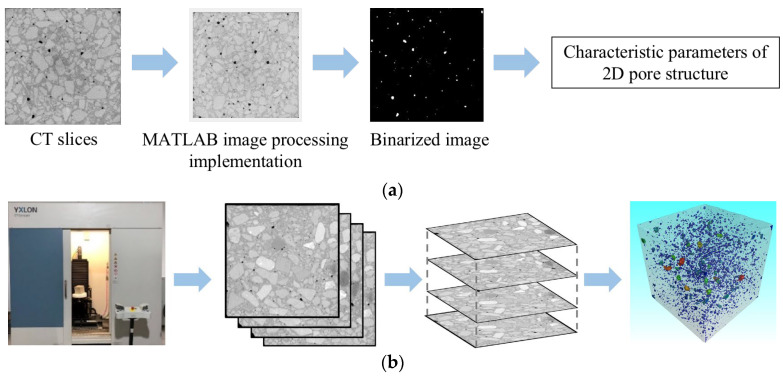
Industrial CT scanning principle. (**a**) Image processing and 2D pore structure extraction; (**b**) 3D reconstruction.

**Figure 3 materials-16-05130-f003:**
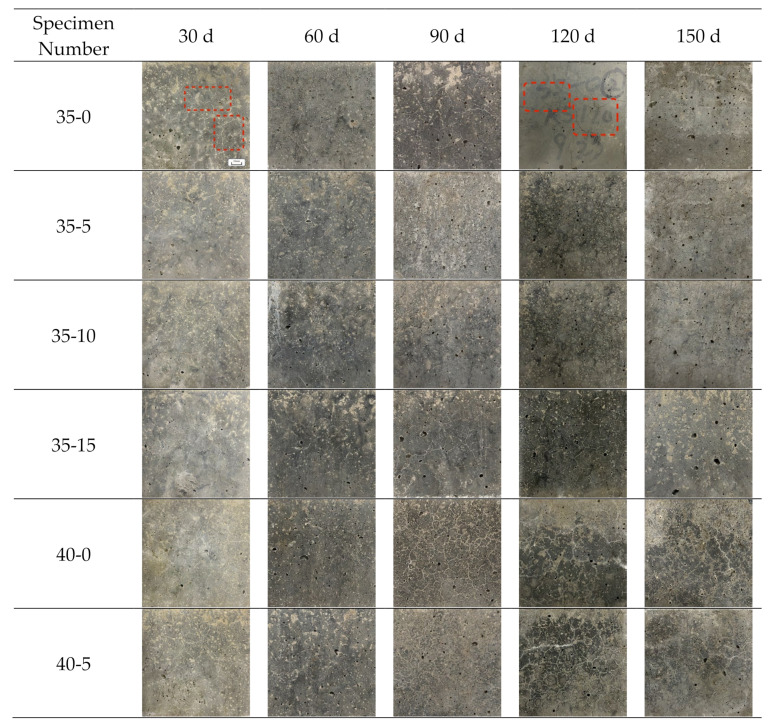
Macroscopic appearance of specimens under sulfate attack for up to 150 days.

**Figure 4 materials-16-05130-f004:**
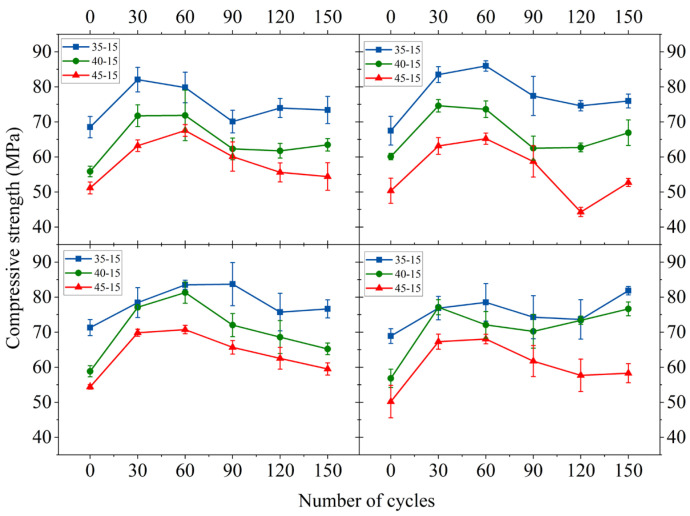
Compressive strength of specimens after sulfate dry–wet cycles.

**Figure 5 materials-16-05130-f005:**
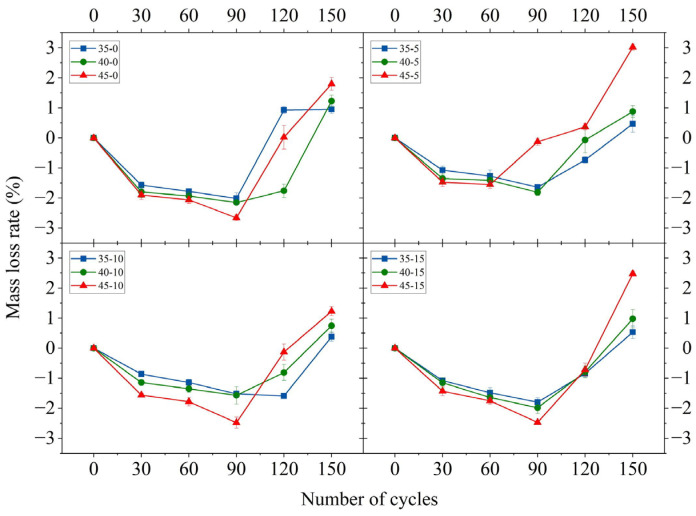
The mass loss rate of specimens after sulfate dry–wet cycles.

**Figure 6 materials-16-05130-f006:**
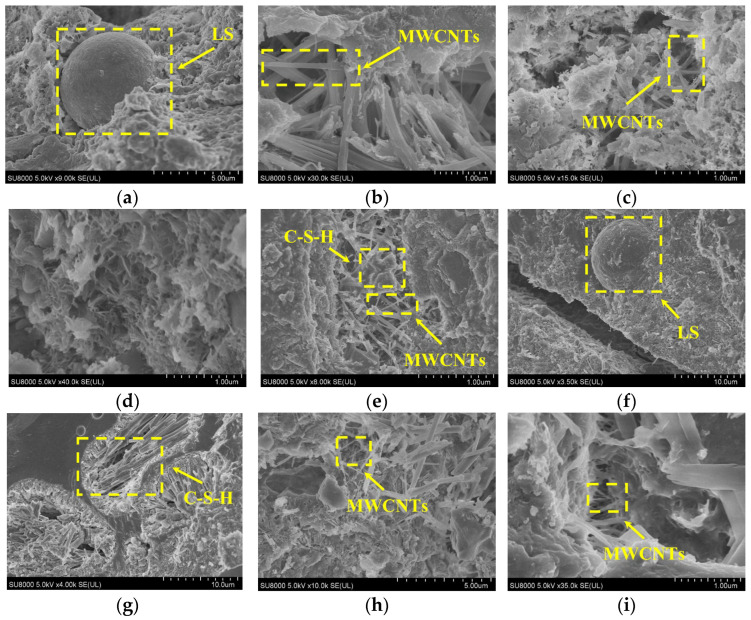
SEM of MWCNT-LSC after sulfate dry–wet cycles: (**a**) 35-0; (**b**) 35-10; (**c**) 45-10; (**d**) 35-0 30 d; (**e**) 35-10 30 d; (**f**) 45-10 30 d; (**g**) 35-0 150 d; (**h**) 35-10 150 d; (**i**) 45-10 150 d.

**Figure 7 materials-16-05130-f007:**
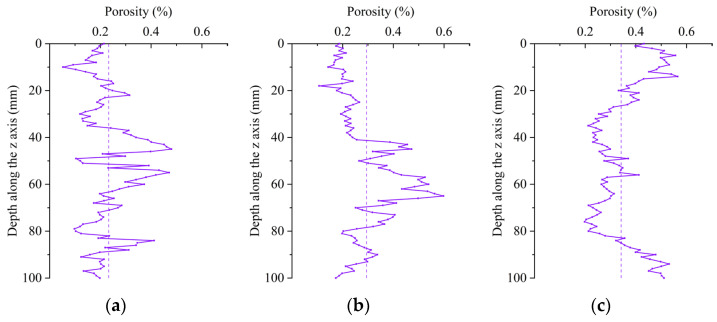
Two-dimensional porosity along the z-axis: (**a**) 35-0; (**b**) 35-0 30 d; (**c**) 35-0 150 d; (**d**) 35-10; (**e**) 35-10 30 d; (**f**) 35-10 150 d; (**g**) 45-10; (**h**) 45-10 30 d; (**i**) 45-10 150 d.

**Figure 8 materials-16-05130-f008:**
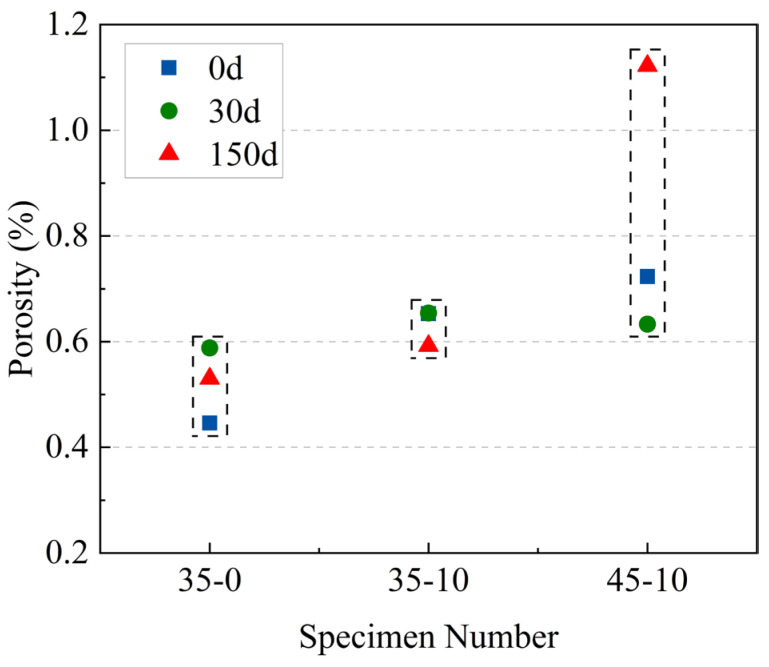
The porosity of 35-0, 35-10, and 45-10 based on CT.

**Figure 9 materials-16-05130-f009:**
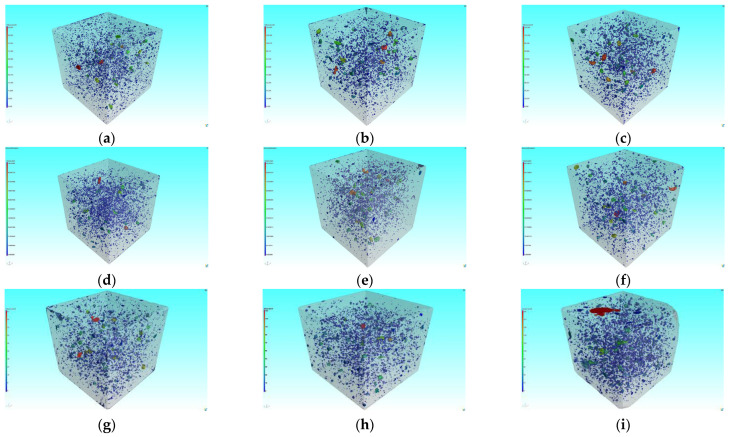
Three-dimensional reconstruction pore structure comparison: (**a**) 35-0; (**b**) 35-10; (**c**) 45-10; (**d**) 35-0 30 d; (**e**) 35-10 30 d; (**f**) 45-10 30 d; (**g**) 35-0 150 d; (**h**) 35-10 150 d; (**i**) 45-10 150 d.

**Figure 10 materials-16-05130-f010:**
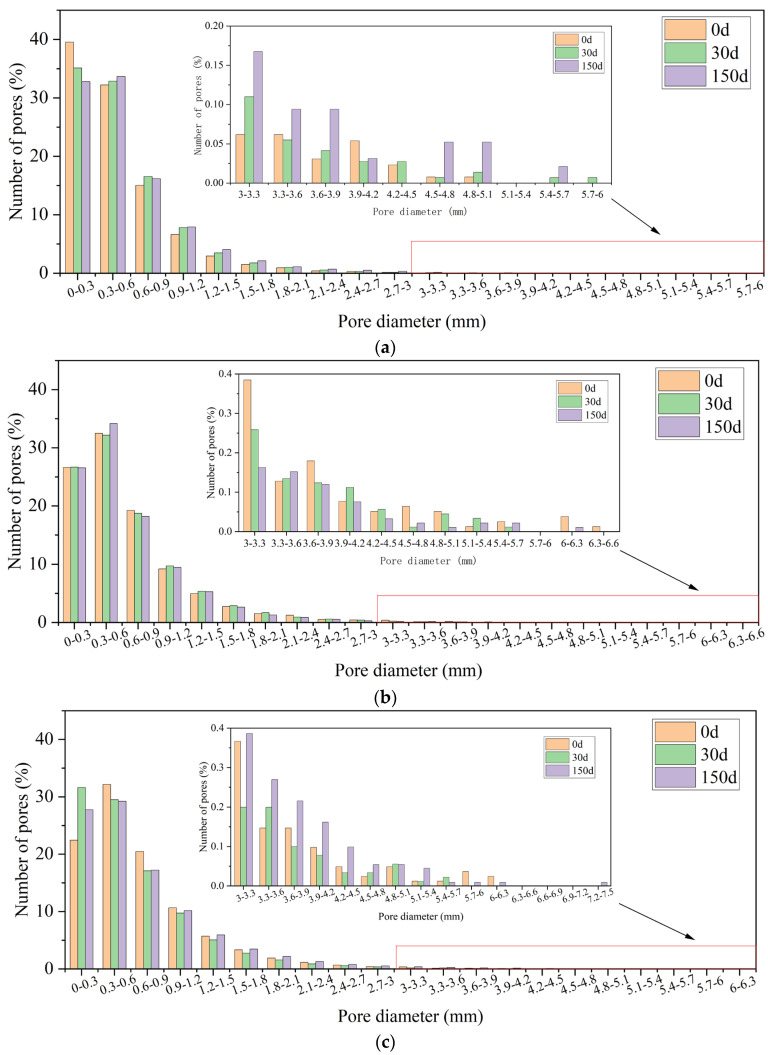
Pore diameter distribution of (**a**) 35-0, (**b**) 35-10, and (**c**) 45-10.

**Figure 11 materials-16-05130-f011:**
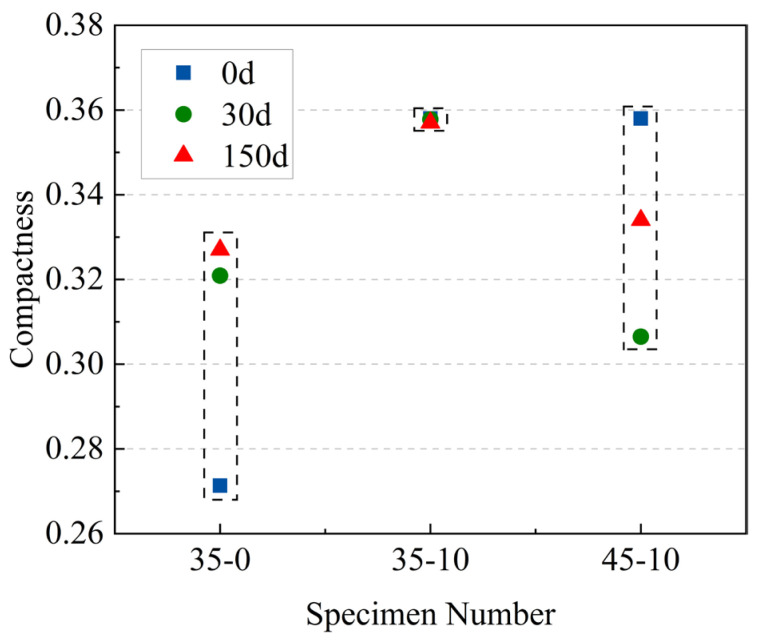
Average compactness of 35-0, 35-10, and 45-10.

**Figure 12 materials-16-05130-f012:**
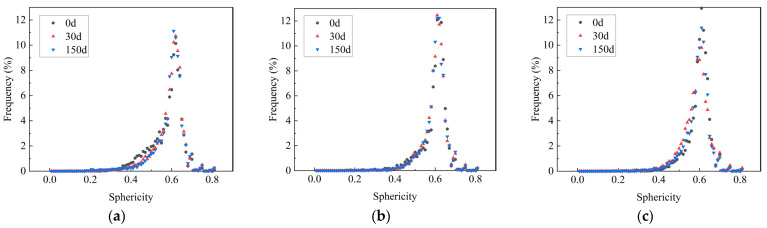
Sphericity analysis of (**a**) 35-0, (**b**) 35-10, and (**c**) 45-10.

**Table 1 materials-16-05130-t001:** Chemical composition of cement and lithium slag (wt./%).

Composition	CaO	SiO_2_	Al_2_O_3_	Fe_2_O_3_	SO_3_	MgO	R_2_O	K_2_O	Na_2_O	P_2_O_5_
Cement	55.32	25.44	7.06	2.89	2.77	2.25	0.88	0.67	0.49	-
LS	22.02	41.72	18.10	1.24	15.14	0.54	-	0.35	0.14	0.37

**Table 2 materials-16-05130-t002:** Properties of MWCNTs used in this study.

Diameter (nm)	Length (µm)	Purity (%)	Ash% (wt.%)	Specific Surface Area (m^2^/g)
40–60	<10	>98	<3	60–100

**Table 3 materials-16-05130-t003:** The mixing ratio of MWCNT-LSC.

Specimen Number	Cement (kg/m^3^)	Water (kg/m^3^)	W/C	Fine Aggregate (kg/m^3^)	Coarse Aggregates (kg/m^3^)	MWCNTs (wt.% OPC)	LS (kg/m^3^)	Water Reducing (%)
35-0	366	160	0.35	774	985	0	91	1.1063
35-5	366	160	0.35	774	985	0.05	91	1.1063
35-10	366	160	0.35	774	985	0.10	91	1.1063
35-15	366	160	0.35	774	985	0.15	91	1.1063
40-0	320	160	0.40	832	977	0	80	0.8800
40-5	320	160	0.40	832	977	0.05	80	0.8800
40-10	320	160	0.40	832	977	0.10	80	0.8800
40-15	320	160	0.40	832	977	0.15	80	0.8800
45-0	284	160	0.45	869	980	0	71	0.7820
45-5	284	160	0.45	869	980	0.05	71	0.7820
45-10	284	160	0.45	869	980	0.10	71	0.7820
45-15	284	160	0.45	869	980	0.15	71	0.7820

**Table 4 materials-16-05130-t004:** CT scanning parameters.

Scanning voltage (kV)	430
Scanning electric current (mA)	1.55
Maximum working power (kW)	0.70
Scanning spacing (mm)	0.50
2D pixel size (mm)	0.127
Enlargement factor	2.02
Focus–detector distance (mm)	1380.28
Focus–specimen distance (mm)	684.60
Integral time (ms)	50
Enlargement factor	Linear array scanning

**Table 5 materials-16-05130-t005:** Pore characteristics of MWCNT-LSC specimens in each scanning section.

Specimen Number	Pore Characteristic	Time (Days)		
		0	30	150
35-0	Average diameter (mm)	0.516	0.553	0.588
	Maximum diameter (mm)	4.838	5.942	5.58
	Average surface area (mm^2^)	2.535	2.834	3.426
35-10	Average diameter (mm)	0.672	0.668	0.646
	Maximum diameter (mm)	6.380	5.624	6.222
	Average surface area (mm^2^)	4.341	4.223	3.800
45-10	Average diameter (mm)	0.713	0.640	0.716
	Maximum diameter (mm)	6.165	5.609	7.346
	Average surface area (mm^2^)	4.681	4.143	5.330

**Table 6 materials-16-05130-t006:** Comparison of pore size distribution.

Specimen Number	Time (Days)	Proportion of Aperture Distribution (%)			
		d < 1 mm	1 mm ≤ d < 3 mm	3 mm ≤ d < 5 mm	d ≥ 5 mm
35-0	0	89.48	10.27	0.25	0
	30	87.91	11.79	0.28	0.02
	150	85.79	13.70	0.49	0.02
35-10	0	81.99	16.99	0.92	0.10
	30	81.73	17.48	0.73	0.06
	150	82.82	16.55	0.58	0.05
45-10	0	79.42	19.62	0.87	0.09
	30	82.18	17.09	0.70	0.03
	150	78.18	20.50	1.23	0.09

## Data Availability

The data used in the article can be obtained from the author.

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
