# Peer review of "Research on Performance Deterioration of Multi-Walled Carbon Nanotube–Lithium Slag Concrete under the Coupling Effect of Sulfate Attack and Dry–Wet Cycles"

_materials, 2023, doi:10.3390/ma16145130_

Round 1

Reviewer 1 Report

For methodology, it is suggested to improve Table 3 by including % of MWCNT. So that easy for the reader to examine the finding.

For result & discussion

- the graph and photo/image are poor. so difficult to do cross checking while reading the discussion. A major revision on the figure is required.

- It is suggested to clearly mark any major changes on Figure 3, which inline with discussion.

- did the reconstruction/method to identify the pore structure is discuss in the methodology. If not, it is suggested to include this in the methodology.

Author Response

The authors thank the reviewer for the valuable and careful comments. All of the following comments have been addressed in the revised version of the manuscript.

Reviewer 2 Report

The paper study the effect on  MWCNT reinforced concrete on coupling dry-wet cycles with sulfate attacks and concludes that adding MWCNT improves durability with respect to sulfate attack under dry-wet cycles. This research complements the state of the art regarding MWCNT reinforced materials. Though there has been lot of work done in the past on these materials, there are a lot of open questions still. After major revisions, this paper should contribute to building a better understanding of these materials. 

The series of tools assembled to characterize finely concrete porosity  are very interesting and the quantity of assembled data is impressive. So this paper has a strong potential. However, in its present state the paper is not convincing. 

The most important issue in the paper is the lack of variability analysis. Only one sample is fabricated by formulation, so there is no way to get insight into the variability in each material. At least for the one or two best formulations identified, several samples (at least 3) should be fabricated to provide information on material variability (which is standard practice in studies of concrete materials). This is a critical issue of this paper considering that MWCNT-reinforced concrete are known to have significant variability. Example in figures 4 and 5: no standard deviation can be provided on the compressive strength data. 

Moreover, for each experiment, the paper fails to provide a synthetic understanding of the outcome of each experiment, jumping instead to (often unsubstantiated by either literature or experiments) explanation. This difficulty to get a synthetic view of the results is reinforced by the absence of an analysis/interpretation section. 

Also, most of the images are extremely blurry and very hard to read even at highest magnification (eg figure 6), which reinforces the hardship in following the analysis of the characterzations. When the images are actually readable, the modalities of exploitation of data remains very unclear. 

Exemple in image 3: the authors illustrate with labels on the image what they consider to be a pore or a crack. Possibly they could provide quantitative analysis of the quantity of pores and cracks instead of providing high-level description of the images. Column 30day, there is a shade on the bottom-left of each image, is it an effect of the lighting or an actual coloring of the samples. 

Same in Figure 7: there is so much spatial variability between two points than at minimum a 3 point moving average is needed to get a synthtic understanding of the porosity pattern.  

Finally, in the bibliography, the paper fails to cite a series of papers on sulfate attacks of MWCNT-reinforced concrete. Only one paper is cited (ref 37), a lot of others are not, for instance  

Repair mortars incorporating multiwalled carbon nanotubes: Shrinkage and sodium sulfate attack; DJ Souza, LY Yamashita, F Dranka… - Journal of Materials in …, 2017 - ascelibrary.org

Grey entropy analysis of strength and void structure of carbon nanotubes concrete under the coupling of sulfate attack and freeze-thaw cycles

X Cheng, W Tian, J Gao, J Guo, X Wang - Construction and Building …, 2022 - Elsevier

Preparation and durability of cement-based composites doped with multi-walled carbon nanotubes. BM Wang, S Liu, Y Han, P Leng - … and Nanotechnology Letters, 2015 - ingentaconnect.com

Effect of the dosage of MWCNT's on deterioration resistant of concrete subjected to combined freeze–thaw cycles and sulfate attack. F Gao, W Tian, Y Wang, F Wang - Structural Concrete, 2021 - Wiley Online Library

English needs to be cross-checked. The use of past and present tenses appears often wrong, some words are missing or sentences are incomplete. 

Author Response

(The authors gave the same response as above.)

Reviewer 3 Report

This paper provides experimental outcomes of the sulfate resistance performance of concrete multi-walled carbon nanotubes (MWCNTs). Numerous experiments were well-designed and conducted to show that the effect of MWCNTs addition to improve the durability, which is very interesting results. To further improve the quality of the paper, please consider the following points;

・        The CNT content is probably expressed as a percentage of the cement mass. Please make this clear in the body text.

・        In 2.3.1 (and others), GB (Chinese standard) are indicated. Because this is an international journal, please indicate the international standard such as ISO. Otherwise, please indicate the correspondence with the ISO (IDT, MOD etc.)

・        It is not clear what is the specific information from the photo of the testing machine in Figure 1. Please explain explicitly what readers can obtain from these exterior photographs. If it is just simple exterior photos, they can be removed.

・        In 3.1, the cracks are evaluated. However, qualitative descriptions such as "more cracks were observed (line 183)" and "fewer cracks than those without MWCNTs (line 185)" are shown. Please make the evaluation as quantitative as possible. From the photos of the specimens shown in Figure 4, there does not appear to be enough difference to be obvious. In addition, please display a scale in Figure 4.

・        The horizontal axis in Figure 5 should be "Number of cycles" not "Time (days)".

・        Regarding the SEM image in Figure 7, the fibrous material shown here seems too thick to be MWCNTs. For example, if the fibrous material shown horizontally in the center of the yellow square in Figure 7(b) is a single fiber, its diameter would be about 300 nm (0.3 μm). If the area enclosed by the vertically long square in Figure 7(h) is a single fiber, its diameter (width) is about 1500 nm (1.5 μm), which is not a nanotube. At the resolution of the PDF file that the reviewer has, it is not possible to determine whether these fibrous objects are made of a single material or many fibers clustered together. The representation should clearly show that these are MWCNTs.

・        The smallest pore size range shown in Figure 11 is 0 to 0.3 mm (300,000 nm), a significant difference from the diameter range that MWCNTs can fulfill. The 3D image shown in Figure 10 would also might have the same range. Please clearly indicate the mechanism the amount of voids of this size has been reduced by MWCNTs.

Author Response

The authors wish to thank the editor and reviewers for their very thorough, insightful, and constructive reviews. All comments have been addressed in the revised version of the manuscript.
